# Symmetrically pulsating bubbles swim in an anisotropic fluid by nematodynamics

Sung-Jo Kim[1,2], Žiga Kos [3,4,5], Eujin Um [1] & Joonwoo Jeong [1]✉

Swimming in low-Reynolds-number fluids requires the breaking of time-reversal symmetry and centrosymmetry. Microswimmers, often with asymmetric shapes, exhibit nonreciprocal motions or exploit nonequilibrium processes to propel. The role of the surrounding fluid has also attracted attention because viscoelastic, non-Newtonian, and anisotropic properties of fluids matter in propulsion efficiency and navigation. Here, we experimentally demonstrate that anisotropic fluids, nematic liquid crystals (NLC), can make a pulsating spherical bubble swim despite its centrosymmetric shape and time-symmetric motion. The NLC breaks the centrosymmetry by a deformed nematic director field with a topological defect accompanying the bubble. The nematodynamics renders the nonreciprocity in the pulsation-induced fluid flow. We also report speed enhancement by confinement and the propulsion of another symmetry-broken bubble dressed by a bent disclination. Our experiments and theory propose another possible mechanism of moving bodies in complex fluids by spatiotemporal symmetry breaking.

Low-Reynolds-number (Re≪1) hydrodynamics governs the locomotion of microswimmers[1,2]. The Navier-Stokes equation becomes time-independent when the viscous force dominates over the inertial force in the low Re regime. Applying this equation to Newtonian incompressible fluids, the scallop theorem implies that swimmers exhibiting only a nonreciprocal motion may gain net propulsion via time-reversal symmetry breaking[1,3]. Examples in the nature include whip- or corkscrew-like flagellar motions and cilia's metachronal wave of microorganisms[4–6]. External field-driven artificial swimmers[5,7–11] and various theoretical models, such as a swimming sheet and a three-link swimmer, mimic the biological motions[5,6,12–15]. It is noteworthy that the symmetry-breaking motions of the swimmers set their swimming direction, i.e., the head and tail.

Microswimmers with no mechanical motions also break the symmetries in various ways[5,16–18]. Self-propelling micro-objects often generate and sustain a gradient, e.g., of chemicals and temperature, over their anisotropic bodies imposing the head and tail. Moreover, the gradient-generating processes, often involving chemical reactions and external energy input, are at nonequilibrium, which inherently breaks the time-reversal symmetry. One well-known example is an active Janus particle with anisotropic chemical or electrical properties in a chemically reactive medium or under external electric fields[16–18]. Even with no intrinsic anisotropy, a spontaneous symmetry breaking can also give rise to net propulsion in active emulsion and Quincke rollers[16,19–21]. It is no surprise that many studies have focused on the understanding and design of symmetry breaking by the microswimmers themselves.

Deploying complex fluids environment is another strategy to achieve symmetry breaking and even guide their swimming direction. For instance, an artificial scallop can swim despite its reciprocal motion exploiting time-asymmetric motions in non-Newtonian fluids[22]. Nematic liquid crystals (NLC), as a structured fluid with elasticity, can accommodate microswimmers. A topologically required point defect accompanies a spherical colloid dispersed in the NLC and breaks the colloid's symmetry, resulting in net propulsion[16,23–26]. Furthermore, an aligned NLC can guide the swimming directions of motile objects in the NLC; flagellated bacteria favor swimming along the alignment direction[15,16,25].

In this study, utilizing the symmetry breaking in a surrounding fluid solely, we experimentally demonstrate a spherical swimmer displaying a time-symmetric size pulsation. Body motion-assisted

¹Department of Physics, Ulsan National Institute of Science and Technology, Ulsan, Republic of Korea. ²Center for Soft and Living Matter, Institute for Basic Science, Ulsan, Republic of Korea. ³Faculty of Mathematics and Physics, University of Ljubljana, Ljubljana, Slovenia. ⁴Jožef Stefan Institute, Ljubljana, Slovenia. ⁵International Institute for Sustainability with Knotted Chiral Meta Matter, Hiroshima University, Higashihiroshima, Japan. ✉e-mail: jjeong@unist.ac.kr

microswimmers hitherto studied should have intrinsic anisotropy and show either nonreciprocal or time-asymmetric motion to gain net propulsion. However, our pulsating spherical bubble in NLC acquires the centrosymmetry breaking by having a point defect, and nematodynamics in the viscoelastic and anisotropic NLC renders time-reversal symmetry breaking despite the time-symmetric pulsation.

## Results and Discussion

We recruit pulsating spherical bubbles dispersed in a homogeneously aligned nematic liquid crystal (NLC) cell as microswimmers (Fig. 1a). Two surface-treated substrates sandwich the NLC to form the homogeneously aligned cell of a uniform thickness (see Methods and Supplementary Fig. 1)[27]. The nematic director **n**, representing the average direction of the rod-like LC molecules, is aligned uniaxially, defining the far-field director **n$_0$** parallel to the substrates. Spherical bubbles of approximately 100 μm in diameter, changing their radii under pressure modulation, are dispersed in the NLC (Fig. 1b). The buoyant bubbles float but do not touch the top substrate because of the elastic repulsion in NLC[27–30]; They remain spherical if they are smaller than the distance between the substrates.

The bubble distorts the homogeneously aligned NLC director field to satisfy the boundary conditions. The directors are perpendicular to the bubble surface[29,31,32], which causes the bubble to acquire a topological defect conserving the zero net topological charge of the homogeneous director configuration[33,34]. Figures 1c and d illustrate director configurations with a disclination ring called the Saturn-ring (SR) and a point defect called the hyperbolic hedgehog (HH), respectively[35]. The bubbles accompanying each defect are labelled SRB and HHB, respectively. The energetics regarding the bubble size and confinement determines the director configuration and the type of accompanying defects[28,36]. For instance, as displayed in Fig. 1e, we can transform SR into HH by decreasing the bubble size (Supplementary Fig. 2). The location of the HH (left or right side of the bubble in Fig. 1a) is determined randomly. This point defect breaks the centrosymmetry, meaning that the defect side of HHB differs from the defect-free side, in contrast to the centrosymmetric SRB.

The pulsating HHB swims toward the accompanying HH, whereas the displacement of pulsating SRB is negligible (Fig. 1c and d). We prepare and investigate a single bubble in the whole sample cell to exclude interference from other bubbles (see Methods for experimental details and Supplementary Movie 1). As shown in Fig. 1b, $\kappa = \frac{t}{T}$ represents the cycle number of the pulsating bubble when $t$ and $T$ are the elapsed time and period of the sinusoidal pressure modulation, respectively. The centrosymmetric SRB practically does not move (Fig. 1c and Supplementary Movie 2); The centre position $z_B$ of SRB moves by approximately 1 μm during $\kappa = 480$ with no change in the bubble average radius $R$. We then decrease the radius of the pulsating SRB gradually by applying the positive DC offset pressure $P_{offset}$ to the sinusoidal pressure modulation (see Methods for experimental details). Figure 1e in the bubble's centre frame reveals the transformation from SRB to HHB as the SR shrinks to HH. Subsequently, the centrosymmetry-broken HHB shown in Fig. 1d and Supplementary Movie 3 swims toward HH along **n$_0$** by the periodic size modulation. Specifically, the HHB of the average radius $R_0 = 60$ μm swims at the average speed of 0.2 μm/s under 4-Hz pulsation with the amplitude $\Delta R \approx 5$ μm. We confirm that this motion is not an overall drift because multiple HHBs in the same cell migrate toward their own HHs (Supplementary Movie 4).

We find the bubble's centre translates while oscillating with a phase delay to the sinusoidal radius oscillation. As shown in Fig. 2a, upon the sinusoidal pressure modulation of the frequency $f$, HHB's radius $R(t)$ oscillates about $R_0$ with the same frequency $f$ and an amplitude $\Delta R$, following the isothermal volume change of the ideal gas (the red solid line in Fig. 2a), which is expressed as $R(t) \approx R_0(1 + \frac{\Delta R}{R_0} \sin 2\pi f t)$ with a linear approximation. This $R(t)$ results in the motion of HHB's centre $z_B(t)$ with an oscillation amplitude $\Delta z$ and a linear translation $z_0(t)$, i.e., $z_B(t) \approx z_0(t) + \Delta z \sin(2\pi f(t - t_d))$ with $z_0(t) = Ut + z_{const}$, a constant velocity $U$, and a constant position $z_{const}$ (Fig. 2a). As shown in the last row of Fig. 1d, we define the positive $z$-direction in the bubble's centre frame as the defect-to-bubble-centre direction parallel to the **n$_0$**. To our interest, $z_B(t)$ has a time delay $t_d$ to $R(t)$, and Fig. 2b indicates the phase-delay $\Psi = 2\pi f t_d \propto f^{0.46 \pm 0.02}$. We can

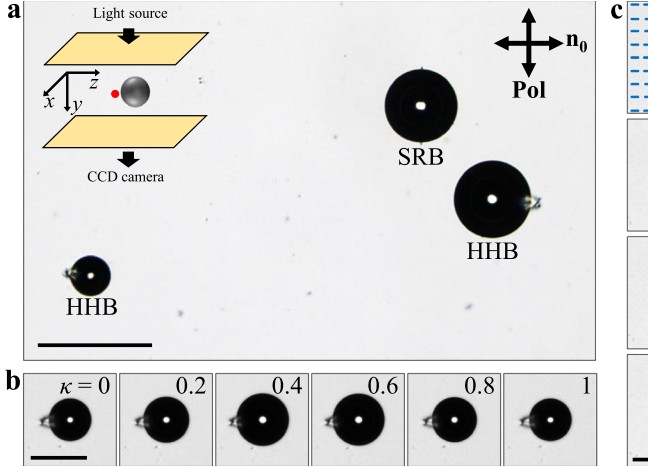

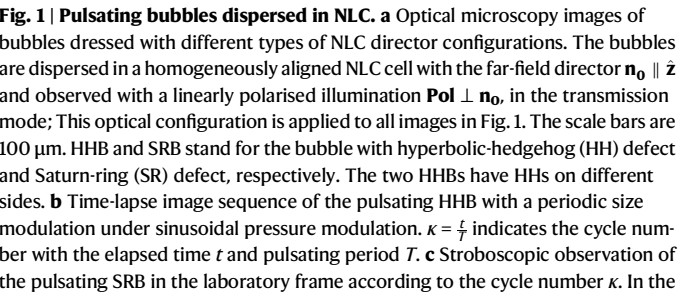

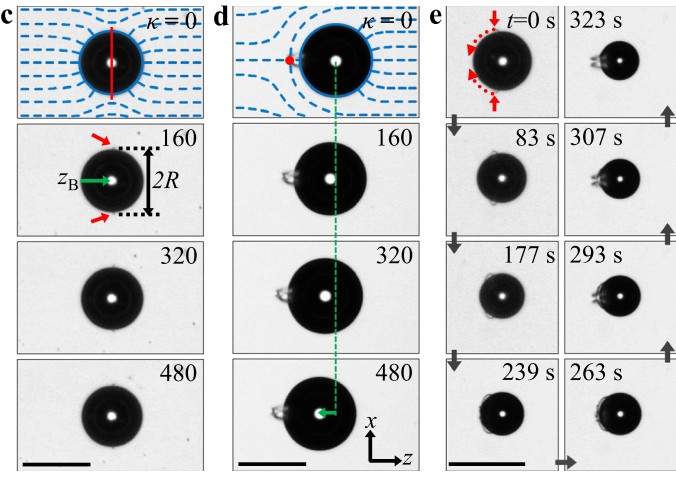

**Fig. 1 | Pulsating bubbles dispersed in NLC. a** Optical microscopy images of bubbles dressed with different types of NLC director configurations. The bubbles are dispersed in a homogeneously aligned NLC cell with the far-field director **n$_0$** ∥ **ẑ** and observed with a linearly polarised illumination **Pol** ⊥ **n$_0$**, in the transmission mode; This optical configuration is applied to all images in Fig. 1. The scale bars are 100 μm. HHB and SRB stand for the bubble with hyperbolic-hedgehog (HH) defect and Saturn-ring (SR) defect, respectively. The two HHBs have HHs on different sides. **b** Time-lapse image sequence of the pulsating HHB with a periodic size modulation under sinusoidal pressure modulation. $\kappa = \frac{t}{T}$ indicates the cycle number with the elapsed time $t$ and pulsating period $T$. **c** Stroboscopic observation of the pulsating SRB in the laboratory frame according to the cycle number $\kappa$. In the

first row, we use a red solid line to indicate the SR and blue dashed lines to illustrate the NLC director configuration. $z_B$, indicated by the green arrow in the second row, is the centre of the bubble of the diameter $2R$, and the red arrows point to the SR. **d** Stroboscopic observation of the pulsating HHB and its translation in the laboratory frame. The red dot in the first row indicates HH in the director configuration (blue dashed lines). The HHB translates toward the HH from the initial position, manifested by the green dashed line and arrow. We define the positive $z$-direction as the direction from HH to the centre of bubble. **e** SR-to-HH transformation in the bubble frame. The red dotted arrows at $t = 0$ s illustrate how the SR collapses into the HH. The bubble gradually shrinks because we apply a positive offset pressure in addition to the sinusoidal pressure modulation.

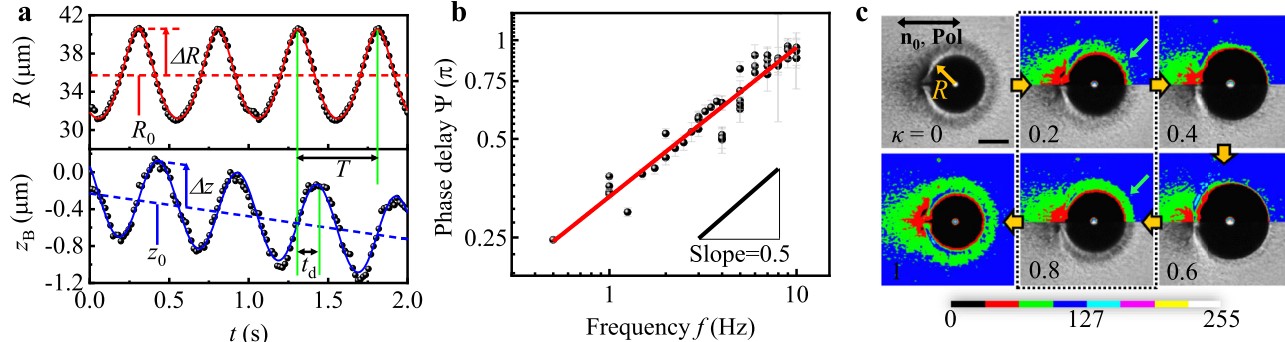

**Fig. 2 | Measurements of the pulsation-induced propulsion of HHB. a** Size and position of a representative pulsating HHB. The radius $R$ oscillates about $R_0$ with the amplitude $\Delta R$ by the sinusoidal pressure modulation of the period $T$. The red solid line corresponds to a fit according to the isothermal volume change of an ideal-gas bubble. The position $z_B(t)$ of the bubble's centre also exhibits an oscillation with the amplitude $\Delta z$ and the same frequency $f = T^{-1}$, but with a linear translation shown as $z_0$ and a time delay $t_d$ to $R(t)$. The blue solid line is the best fit with the oscillation and linear translation. **b** Scaling relation between a phase delay $\Psi = 2\pi f t_d$ and $f$. Each data point is the average value from a 2-min-long recorded movie at 60 frames per second, and the error bars represent its standard deviation. The fit line indicates $\Psi \propto f^{0.46\pm0.02}$. **c** Polarised optical microscopy observations of HHB during a single pulsation cycle. With the incident polarisation **Pol** parallel to the far-field director **$n_0$**, we observe the transmitted light intensity around the pulsating HHB according to the cycle number $\kappa$ with a discrete colour mapping of 8-bit intensity. The scale bar is 50 μm.

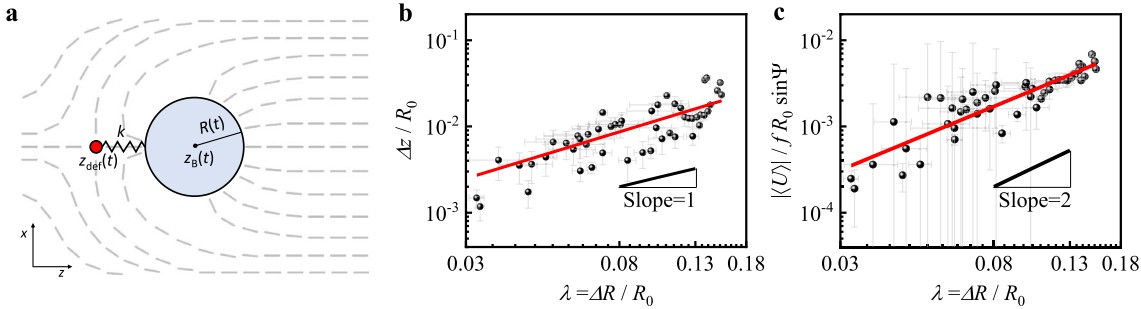

**Fig. 3 | An oscillating dumbbell model for the pulsating HHB and its comparison with experimental data. a** A schematic diagram of a pulsating HHB as an oscillating dumbbell. The pulsating HHB of the time-varying radius $R(t)$ with its centre at $z_B(t)$ accompanies a point defect at $z_{def}(t)$. The connecting spring of a spring constant $k$ represents an effective quadratic potential between the defect and the bubble. Dashed lines sketch the deformed nematic director field with the perpendicular anchoring at the bubble surface. **b** and **c** Scaling relation between the dimensionless oscillation amplitude $\frac{\Delta z}{R_0}$, dimensionless swimming speed $\frac{|\langle U\rangle|}{fR_0\sin\Psi}$, and deformation ratio $\lambda$, where $\Psi = 2\pi f t_d$ is the phase delay. The fit lines in **b** and **c** denote that the oscillation amplitude is proportional to $\lambda^{1.3\pm0.1}$, and the dimensionless speed is proportional to $\lambda^{1.7\pm0.1}$. The data in Fig. 2b, Figs. 3b and c are from the same dataset of observed pulsating HHBs confined in a 155 μm-thick cell. Each data point is the average value from a 2-min-long recorded movie at 60 frames per second, and the error bars represent its standard deviation; The large errors in **c** partly result from the error propagation of $\sin\Psi$ in the denominator.

exclude possible roles of inertia[37] and NLC's shear-rate dependent viscosity[22] in HHB's propulsion. The density (≈1.0 g/cm³), viscosity (≈$10^{-1}$ Pa·s), characteristic time and length scales ($T \approx 1$ s and $\Delta R \approx 10$ μm) and flow speed ($\frac{dR}{dt}$) give Re<$10^{-4}$ and shear-rate <1 s⁻¹, where our NLC shows negligible shear-rate dependent viscosity[38].

The comparison of the two time scales hints that nematodynamics around the pulsating HHB results in the experimentally observed phase delay and time-reversal symmetry breaking, enabling the swimming motion. The nematic director field at a length scale $l$ has a diffusive elastic response with a timescale of $\tau = \gamma_1 l^2/K$; $\gamma_1$ is the nematic rotational viscosity, and $K$ is the nematic elastic constant in the one-constant approximation[39]. Within the slow-pulsation regime where the director oscillation period $T$ by the pulsation is much longer than $\tau$, the nematic directors have enough time to globally adapt to the oscillating environments, resulting in negligible time-reversal symmetry breaking and no net translation according to the Scallop theorem. However, for the fast-pulsation regime where $T$ is comparable or shorter than $\tau$, the directors cannot respond quickly enough in the quasi-static way, causing that the directors near the pulsating HHB may change in a time-asymmetric manner with a time delay to the fast pulsation.

The elasticity-mediated director dynamics resulting from a local director oscillation can be illustrated in the following one-dimensional system simplifying the director fields around a pulsating bubble. Nematodynamics formulates that the director tilt angle $\phi(x, t)$ at time $t$ and distance $x$ from a surface obeys the dynamic equation $\gamma_1 \frac{\partial\phi}{\partial t} = K \frac{\partial^2\phi}{\partial x^2}$[39]. With a periodic distortion $\phi(x = 0, t) = \phi_0 \sin(\omega t)$ imposed at $x = 0$, the time-dependent solution for the tilt angle is $\phi(x, t) = \phi_0 e^{-x/\zeta} \sin(\omega t - x/\zeta)$, where $\zeta = \sqrt{\frac{2K}{\omega\gamma_1}}$. The elastic diffusion of the oscillating director decays exponentially away from the surface with a characteristic length scale $\zeta$. Additionally, to our interest, the phase delay term scales as $\sqrt{\omega}$, and a similar scaling is experimentally observed in the phase delay $\Psi$ between $R(t)$ and $z_B(t)$ as shown in Fig. 2b: $\Psi \propto f^{0.46\pm0.02}$. This phase delay breaks the time-reversal symmetry, making the nematic director fields around the expanding and shrinking bubble different. We find experimentally no clear $\lambda$-dependency of the phase delay under the same frequency as shown in Supplementary Fig. 3. For typical material parameters, such as $K \approx 10$pN and $\gamma_1 \approx 0.1$ Pa·s, the characteristic length scale $\zeta$ at $f = 2$ Hz is approximately 4 μm - $0.1 R_0$. This indicates that the time-asymmetric director deformation in the vicinity of the bubble, which includes the point-defect region, should be mostly responsible for the net swimming motion.

We experimentally verify the time-reversal symmetry breaking of the director fields around the pulsating HHB. As depicted in Fig. 2c and

Supplementary Movie 5, we observe the NLC around a pulsating HHB using polarised optical microscopy and measure the transmitted intensity profile. Because the transmitted intensity of polarised light through the birefringent NLC reflects the director configuration along the beam path[40], the time-asymmetric intensity profile reveals that the director configurations near the HHB during the expansion and shrinkage differ. For instance, as shown in Fig. 2c, the HHB has the same size, i.e., $R(\kappa = 0.2) = R(\kappa = 0.8)$, but the transmitted intensity profiles near the bubble do not overlap; See the area and location of the green equi-intensity region indicated by green arrows. Namely, the sinusoidal pulsation is reciprocal and time-symmetric, but the NLC environment is not.

Here we present an analytical model to explain the pulsating HHB's motion considering the nematodynamics around the bubble accompanying the point defect. As the first approximation, our model considers the HHB in an infinite bulk system with no wall and buoyancy. Then, we characterize the system with a dimensionless Ericksen number $\mathrm{Er} = \frac{\omega \gamma_1 R_0^2}{K}$ that compares a time period of radius oscillation with the nematic director relaxation time at the length scale of $R_0$. For a slow pulsation, i.e., $\mathrm{Er} \ll 1$, the energetics of the nematic director field and the viscous drag on the point defect govern the bubble displacement dynamics (see Methods for the calculation of viscous loss in a pulsating flow). The director field around a spherical bubble with the point defect can be characterized by a distance between the defect and the bubble surface, and the equilibrium distance is determined by a quadratic potential[41]. When the defect deviates from its equilibrium position, a pair of elastic forces aims to restore the equilibrium configuration and displaces the defect and the bubble. The Ericksen stress tensor $\sigma_{ij}^{\mathrm{Er}} = -\frac{\delta f}{\delta \partial_j n_k}\partial_i n_k + f\delta_{ij}$ with the free energy density $f$ of the nematic director field $\mathbf{n}$ formally mediates the forces. However, instead of working directly with the stress tensor, we adopt a coarse-grained approach where the energetics and drag of defect structures determines their dynamics. This is a common approach formulating analytical descriptions of nematodynamics[42].

As shown in Fig. 3a, an effective dumbbell-like model for the defect and the bubble connected by a spring describes a quadratic potential $\mathcal{F} = \frac{k}{2}(d - \epsilon R)^2$, where $d$ is a bubble surface-to-defect distance with the constant $\epsilon = 0.17$ and $k = 16.5\pi K/R(t) = k_0 \frac{R_0}{R(t)}$ is the effective spring constant[41]. Employing the sinusoidal pulsation $R(t) = R_0 + \Delta R \sin \omega t = R_0(1 + \lambda \sin \omega t)$ with the pulsation ratio $\lambda = \Delta R/R$, we find that the spring constant $k$ and the equilibrium length $\epsilon R$ have a first order correction in $\lambda$. The spring transmits a force $\vec{F}$ that drives the overdamped motion of the point defect and the bubble. The drag coefficient of the point defect $c_{\mathrm{def}} = \pi^2 \gamma_1$ is derived in Eq. (13) (see Methods for the calculation of viscous loss in a pulsating flow), where $\gamma_1$ is the rotational viscosity, and the drag coefficient of the gaseous bubble equals $c_B \approx 4\pi\eta$ for an average isotropic viscosity $\eta$[43].

We first consider the slow propulsion regime of $\mathrm{Er} \ll 1$. The oscillation of the bubble radius generates a propulsion force on the bubble (see Methods for derivations)

$$F_{\mathrm{slow}} = \frac{\omega \Delta R R_0 c_B (1+\epsilon)}{1 + \frac{c_B}{c_{\mathrm{def}}}} \cos \omega t. \tag{1}$$

The force is proportional to $\dot{R}(t)$ with the proportionality constant depending on the parameters of the dumbbell-like model. The propulsion force induces a periodic oscillation of the bubble position $z_B$

$$z_B(t) = z_0 + \frac{\Delta R(1+\epsilon)}{1 + \frac{c_B}{c_{\mathrm{def}}}} \sin \omega t. \tag{2}$$

The bubble position $z_B(t)$ in Eq. (2) exhibits no net displacement but a periodic motion in phase with the bubble radius $R(t) = R_0 + \Delta R \sin \omega t$. This is consistent with the Scallop theorem[44], since the pulsation with the repeating expansion and shrinkage is reciprocal.

Now, extending the slow-pulsation model into the fast-pulsation one, we present a minimal model explaining the bubble's net propulsion. As discussed above for the one-dimensional director dynamics model with an oscillating-director boundary condition at finite Ericksen numbers, the nematic response to periodic modulation is not instantaneous. Thus, to construct a minimal model of bubble propulsion in this fast-pulsation regime, we employ a phase delay $\psi$ with respect to $\dot{R}(t)$ in the periodic sinusoidal propulsion force:

$$F_{\mathrm{fast}} = a\omega\lambda R_0^2 \cos(\omega t - \psi), \tag{3}$$

where $a$ is the proportionality coefficient that can depend on the system size and material parameters. At the low Re regime showing the overdamped motion, the propulsion force is counteracted by the Stokes drag on the bubble

$$F_{\mathrm{drag}} = c_B R(t)\dot{z}_B(t) = c_B R_0(1 + \lambda R \sin \omega t)\dot{z}_B(t). \tag{4}$$

Note that, in the view of building a minimal model, we adopt only the sinusoidal propulsion force in Eq. (3) and the sinusoidal drag coefficient in Eq. (4), although higher Fourier modes are possible. As shown in the bottom panel of Fig. 2a, deviations from sinusoidal oscillations are negligible in the experiments, which supports our approximation.

Equating $F_{\mathrm{fast}} = F_{\mathrm{drag}}$ gives the bubble velocity $\dot{z}_B(t)$ with both the oscillation and the translation, which supports our experimental observation. When expanded for $\lambda \ll 1$,

$$\dot{z}_B(t) = \frac{a\omega R_0}{c_B}\left[\lambda\cos(\omega t - \psi) - \lambda^2\cos(\omega t - \psi)\sin(\omega t)\right] + \mathcal{O}(\lambda^3). \tag{5}$$

We integrate the velocity over time to obtain the bubble position

$$z_B(t) = z_{\mathrm{const}} + \frac{a\omega R_0}{c_B}\left[\frac{\lambda}{\omega}\sin(\omega t - \psi) + \frac{\lambda^2}{4\omega}\cos(2\omega t - \psi) - \frac{\lambda^2}{2}t\sin\psi\right]$$
$$+ \mathcal{O}(\lambda^3). \tag{6}$$

The oscillation amplitude $\Delta z$ of $z_B(t) = z_0(t) + \Delta z\sin(\omega t - \psi)$ corresponds to $\lambda\frac{aR_0}{c_B}$ which scales linearly with $\lambda$ and supports the experimentally observed scaling in Fig. 3b, showing the oscillation ratio $\frac{\Delta z}{R_0} \propto \lambda^{1.3 \pm 0.1}$. Note that the additional oscillation contribution with a doubled frequency $2\omega$ should have a minor effect on the bubble position compared to the $\omega$ oscillation term, because of the prefactor of $\lambda^2/4$ with the experimental $\lambda < 0.18$. Importantly, the phase delay $\psi$ results in the net translation term, $-\frac{\lambda^2}{2}t\sin\psi$, giving the time-averaged swimming speed $\langle U \rangle = -\lambda^2 R_0\omega\sin\psi\frac{a}{2c_B}$. This result is in line with the experimentally observed scaling of the swimming speed shown in Fig. 3c, where the dimensionless translation swimming speed $\frac{|\langle U \rangle|}{fR_0\sin\Psi}$ is proportional to $\lambda^{1.7 \pm 0.1}$. The proportional relation between net displacement ($|\langle U \rangle|T$) and oscillation amplitude ($\Delta z$) shown in Supplementary Fig. 4 also support our model shown as Eq. (6). However, our experimental setup limits the ranges of $R_0$, $\lambda$, and $f$; see Methods for the details. Additionally, the data points in Fig. 3b and c have only the limited range of $R_0$, from 27.1 to 37.6 μm, because we want to exclude a confinement effect that will be discussed in the following paragraph. Thus, Fig. 3b and c do not validate the $|\langle U \rangle| \propto R_0$ experimentally.

The origin of the phase delay $\psi$ in Eq. (3), resulting in the net translation of the bubble, deserves further discussion. The one-dimensional director dynamics model explained above shows that the director oscillation decays out exponentially from the boundary. This indicates that the phase-delayed director deformation only in the vicinity of the bubble, including the point defect − matters. Moreover, the force due to a moving point defect is dominant, as shown in Methods. Therefore, in the coarse-grained dumbbell model, we

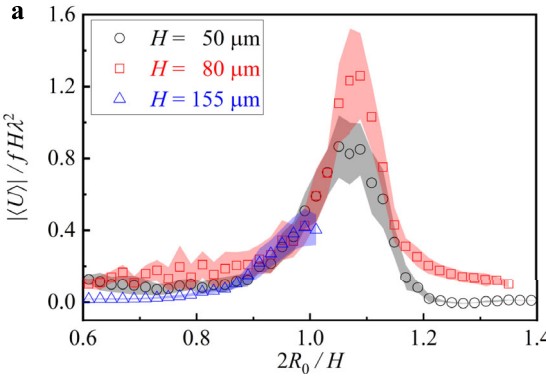

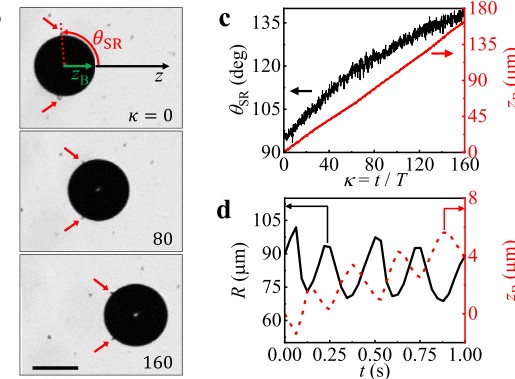

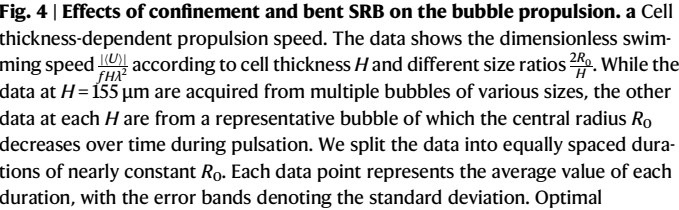

**Fig. 4 | Effects of confinement and bent SRB on the bubble propulsion. a** Cell thickness-dependent propulsion speed. The data shows the dimensionless swimming speed $\frac{|\langle U\rangle|}{fH\lambda^2}$ according to cell thickness $H$ and different size ratios $\frac{2R_0}{H}$. While the data at $H=155$ μm are acquired from multiple bubbles of various sizes, the other data at each $H$ are from a representative bubble of which the central radius $R_0$ decreases over time during pulsation. We split the data into equally spaced durations of nearly constant $R_0$. Each data point represents the average value of each duration, with the error bands denoting the standard deviation. Optimal propulsion is achieved at $\frac{2R_0}{H}\approx1.1$, regardless of $H$. **b** Stroboscopic observation of the pulsating SRB with a bent SR according to the cycle number $\kappa$. The SR pointed by the red arrows does not lie at the equator of the SRB. The angle $\theta_{SR}$ is the angle between the bent SR and the moving direction ($z$-axis). The scale bar is 100 μm. **c** Representative data of a SRB's centre position $z_B$ and $\theta_{SR}$ as functions of $\kappa$. **d** Representative data of the radius $R(t)$ (black solid) and the centre position $z_B(t)$ (red dashed) of the pulsating SRB according to the time $t$.

consider the force by the point defect exhibiting the delayed motion with a finite $\psi$, but do not assume any specific dependence of the phase delay on the model parameters. In Fig. 2(b), experimental data shows that the phase delay $\Phi$ between $R(t)$ and $z_B(t)$ scales as $f^{0.46\pm0.02}$, which is in line with $\psi$ in the one-dimensional model of director oscillation that is proportional to $\sqrt{f\gamma_1/K}$. The dependence of the phase delay on the other material parameters can be tested in future works using different LC materials.

We also discover that an optimal confinement exists for pulsating HHB's propulsion, and the HHB can reach a maximum speed of approximately 1 μm/s, which is about one order of magnitude faster than the slowest observed bubble. The swimming speed of the bubble in Fig. 1d and Supplementary Movie 3 is relatively slow compared to other microswimmers[5], only achieving 0.2 μm/s and meaning $\frac{|\langle U\rangle|T}{2R_0}\sim0.003$. However, regardless of the cell thickness $H$, the dimensionless swimming speed $\frac{|\langle U\rangle|}{fH\lambda^2}$ increases considerably as the bubble diameter $2R_0$ approaches the cell thickness $H$, achieving the maximum value near $\frac{2R_0}{H}\approx1.1$, as shown in Fig. 4a. Because the spherical HHB should be squeezed when the diameter reaches $\frac{2R_0}{H}\approx1$, the bubble's shape in this range keeps transforming between the sphere and disk by pulsation. We presume that this shape change enhances the time asymmetry of the pulsation process; The bubble feels a different environment during the expansion and shrinkage, respectively. Supplementary Fig. 5 and Movies 6 and 7 show that flow fields around the bubble indeed change when the spherical bubble becomes the disk. The detailed mechanism of the speed enhancement deserves further investigation.

Lastly, we report that the pulsating SRB also can swim when the bent SR breaks the symmetry[35], as shown in Fig. 4b and Supplementary Movie 8. SRB can retain its bent SR, i.e., not at the equator, possibly because of the cell boundary condition (Supplementary Fig. 2). Figure 4c shows how the bent angle $\theta_{SR}$ and centre position $z_B$ change upon the pulsation cycle. In contrast to the symmetric SRB with its SR at the equator (Fig. 1c), asymmetric SRB realises swimming through pulsation, and the translational and oscillatory motions are similar to those of the HHB (Fig. 4c). We find no strong correlation between the translational speed and $\theta_{SR}$. The $\theta_{SR}$ changes spontaneously during the pulsation cycles. This observation demonstrates that centrosymmetry breaking in any form can lead to net propulsion when combined with the time-reversal-symmetry-breaking NLC relaxation.

The main quest for propulsion in a low Re environment is to break the symmetries. This work demonstrates that even a symmetric object

exhibiting time-symmetric motion can swim by symmetry breaking solely in a structured fluid. Our findings could help us better understand and design microswimmers, from bacteria to artificial sperm, to navigate complex environments. Specifically, a relaxation in complex fluids, responsible for time-reversal symmetry breaking in our case, could be exploited to increase swimming efficiency. Moreover, the observed existence of optimal confinement for propulsion may shed light on the unexpected roles of confinement, e.g., speed enhancement. Lastly, beyond the single-swimmer behaviour studied here, collective swimming resulting from the interaction and symmetry-breaking in complex fluids would be an intriguing question to pursue.

## Methods
### Materials, sample preparation, and optical microscopy
We conduct all experiments using 4-cyano-4'-pentylbiphenyl (5CB, Sigma-Aldrich) as the anisotropic viscoelastic medium at the room temperature 22 ± 2 °C, at which 5CB has the nematic phase. This 5CB is practically incompressible because its volume change ratio is only $10^{-6}$ when the pressure increase by 0.5 MPa from the ambient pressure[45].

A sample cell with a single bubble can be prepared in three steps. First, we prepare an empty sandwich cell with two parallel polyimide-coated glass substrates[27]. They are rubbed along the same direction and assembled face-to-face to impose a uniaxial planar alignment of 5CB at the surface, as displayed in Supplementary Fig. 1a; the NLC directors align along the rubbing direction. The cell gap between two substrates is controlled by film spacers of thickness 50, 80, and 155 μm, and the cell area approximately 1 cm × 1 cm. Only two opposite sides of the square cell are sealed with spacers and adhesive to facilitate pressure propagation to the dispersed bubbles through the openings.

In the second step, we fill the sandwich cell with the bubble-dispersed 5CB. Air bubbles are dispersed into 5CB in a vial by bubbling 5CB with a syringe needle, and the bubble volume fraction is controlled by varying the injection volume and speed. Subsequently, we fill the bubble-injected 5CB into the sandwich cell along the rubbing direction through the unsealed sides. Multiple bubbles exist immediately after filling the cell (Fig. 1a). The bubbles float to the top substrate because of buoyancy but make no physical contact with it because of elastic repulsion in the NLC[27–29,31,32].

Finally, we place the homogeneously aligned NLC cell with multiple bubbles in a custom pressure chamber, as shown in Supplementary Fig. 1a. The pressure chamber with the window allows the optical observation of the bubbles and is connected to a pressure

controller (OB1 MK3, Elveflow) that controls the chamber pressure in the range of $|\Delta P| < 0.5$ MPa from the ambient pressure $P_0$. We decrease the radii of the dispersed bubbles by applying the positive DC offset pressure $P_{offset}$ in the pressure chamber, as shown in Supplementary Movie 1. Monitoring this shrinking process, we eliminate all but a single bubble in the whole cell to investigate the dynamics of a single bubble without interference from the other bubbles. The radius of a single bubble can be controlled by applying pressure. For example, we increase the radius of the small HHB in Fig. 1d after the SR-to-HH transformation to produce a large HHB (Fig. 1e) by applying negative $P_{offset}$.

We use transmission light microscopy with polarised illumination to observe the bubble, as shown in Supplementary Fig. 1b. An inverted microscope (IX73, Olympus) with a 4 × and 10 × objective lenses and a CCD camera (STC-MC202USB, Omron Sentech) captures the motion of the bubble at a maximum acquisition rate of 60 frames per second. The polarised illumination is derived from the linear polariser **Pol** placed in front of the halogen lamp. When the **Pol** of the illumination is perpendicular to the far-field director $n_0$ (**Pol** $\perp$ $n_0$), the boundary of the bubble can be clearly identified, as shown in Figs. 1 and 4b. When **Pol** $\parallel$ $n_0$, as shown in Fig. 2c, the transmitted light intensity reflects the non-uniform director field[40], which allows us to observe qualitatively how the director configuration responds to the pulsation of the bubble.

## Size modulation of the spherical bubble and its measurement

We modulate the size of the bubble by controlling the pressure in the chamber. The bubble remains spherical because of the dominant surface energy with the surface tension $\sigma \sim 10^{-2}$ N/m[30]. For instance, when a bubble of radius $R = 50$ μm pulsates under the pressure modulation of infrasound frequency ($f < 20$ Hz), the surface energy ($\sigma R^2 \sim 2.5 \times 10^{-10}$ J) surpasses both the elastic energy ($KR \sim 5 \times 10^{-16}$ J) and viscous energy ($\gamma f R^3 \sim 2.5 \times 10^{-13}$ J) with the average elastic constant ($K \sim 10^{-11}$ N)[46] and viscosity ($\gamma \sim 10^{-1}$ Pa·s)[40] of 5CB.

The bubble size oscillates almost sinusoidally upon sinusoidal pressure modulation. The infrasound frequency ($f < 20$ Hz) of a wavelength considerably longer than the sample cell size results in uniform pressure across the entire cell. This simplifies the Rayleigh-Plesset equation, describing the dynamics of a spherical bubble in an incompressible fluid into the Young-Laplace equation $P_{bubble} = P_{out} + \frac{2\sigma}{R}$. Since the large bubble size ($R \sim 50$ μm) makes the Laplace pressure $\frac{2\sigma}{R}$ sufficiently smaller than the applied pressure $P_{out}$ with the surface tension $\sigma \sim 10^{-2}$ N/m[30], $P_{bubble} \approx P_{out}$. The pressure $P_{out}(t)$ is $P_0 + P_{offset} - \Delta P \sin 2\pi f t$ consisting of the ambient pressure $P_0$, DC offset pressure $P_{offset}$, and sinusoidally modulating pressure with the amplitude $\Delta P$ and frequency $f$. Applying the isothermal volume change of the ideal gas under $P_{out}(t)$, we determine that the radius $R(t)$ follows $R(t) = R_0 \left( \frac{P_{out}(0)}{P_{out}(t)} \right)^{1/3}$, as displayed by the red solid line in Fig. 2a. The $R(t)$ can be linearly approximated to $R(t) \approx R_0 \left( 1 + \frac{1}{3} \frac{\Delta P \sin 2\pi f t}{P_0 + P_{offset}} \right)$ when $|\frac{\Delta P}{P_0 + P_{offset}}| \ll 1$, and becomes $R(t) \approx R_0(1 + \lambda \sin 2\pi f t)$ with the deformation ratio $\lambda = \frac{\Delta R}{R_0}$ and pulsating amplitude $\Delta R$. We experimentally confirm the proportional relationship between $\lambda$ and $\Delta P$, as shown in Supplementary Fig. 3.

We find the envelopes of the oscillating data under sinusoidal pressure modulation, i.e., the bubble's radius $R(t)$ and the centre position $z_B(t)$, to estimate their oscillation centre and amplitude. As shown in Supplementary Fig. 6, we apply the Envelope method provided by OriginPro 2020 (OriginLab) to determine the enveloping curves connecting the extrema of the oscillating data, e.g., $R_{max}(t)$ and $R_{Min}(t)$. Then, we acquire the oscillation centre $R_0 = \frac{R_{max}(t) + R_{Min}(t)}{2}$ and amplitude $\Delta R = \frac{R_{max}(t) - R_{Min}(t)}{2}$ as functions of time. $R_0$ and $\Delta R$ may change

even under the constant pressure modulation amplitude $\Delta P$ with $P_{offset} = 0$ because 5CB has finite gas solubility[32]. However, the deformation ratio $\lambda = \frac{R_{max}(t) - R_{Min}(t)}{R_{max}(t) + R_{Min}(t)}$ remains constant, and we experimentally confirm it. We apply the same method to retrieve $z_0(t)$ and $\Delta z(t)$ from $z_B(t)$.

Our experiment studies the scaling behavior within the limited range, as shown in Figs. 2 and 3, because of unavoidable experimental limitations and the system's nature. First, optical resolution and the pressure range limit the pulsation ratio $\lambda$. Very small $\lambda$ results in optically unresolvable oscillation amplitude and net displacement of the bubble; The typical net displacement observed after one pulsation cycle with no confinement effect is already sub-micron. On the other hand, because of the inverse relationship between the pressure and the volume = length[3], approximately ten times higher amplitude of pressure modulation than the current value is required to increase $\lambda$ range by the factor of two; We use the maximum pressure range covered by our pressure pump, i.e., $\pm 1$ bar.

In a similar vein, the ranges of frequency and bubble size are limited. It is challenging to observe small bubbles because the small ones dissolve quickly into the LC. The large spherical bubbles demand a homogeneously aligned thick LC cell of mm thickness, which is practically impossible to prepare because of the very long relaxation time. Moreover, as shown in Fig. 4, the propulsion is sensitive to the confinement, i.e., $2R_0/H$; thus, in Fig. 3, we investigate the bubbles of similar radii to exclude the confinement effect in understanding the swimming mechanism. In the case of frequency, the pressure pump's response time of ~ 100 ms sets the maximum frequency ~ 10 Hz. In other words, when the pulsation frequency exceeds 10 Hz, the pumps cannot follow the set frequency and fail to generate the sinusoidal pressure modulation; $\lambda$ decreases at a higher frequency, as shown in Supplementary Fig. 3.

## Viscous loss in pulsating flow

We estimate the magnitude of two propulsion mechanisms of pulsating bubbles: (i) anisotropic viscosity of a dipolar director field structure and (ii) drag force of a moving point defect in the director field. Pulsating bubble generates a radial flow that is subject to the anisotropic viscosity of the surrounding nematic liquid crystal, described by the nematic viscous stress tensor[42]

$$\sigma_{ij}^{viscous} = \alpha_1 n_i n_j n_k n_l A_{kl} + \alpha_2 n_j N_i + \alpha_3 n_i N_j + \alpha_4 A_{ij} + \alpha_5 n_j n_k A_{ik} + \alpha_6 n_i n_k A_{jk}, \tag{7}$$

where $\alpha_i$ are Leslie viscosity coefficients, $A_{ij} = (\partial_i v_j + \partial_j v_i)/2$ is the symmetric shear tensor, and $N_i = \dot{n}_i - ((\nabla \times \vec{v}) \times \vec{n})_i/2$ is the corrotational time derivative of the director. Dipolar director structure around the bubble breaks the symmetry and allows for a net force due to the bubble radial expansion. To estimate the propulsion force at Er $\ll 1$, we take a stationary dipolar director field ansatz[41]

$$\vec{n}(\vec{r}) = \left( \frac{R_0^2}{r^3} x, \frac{R_0^2}{r^3} y, \sqrt{1 - \frac{R_0^4}{r^6}(x^2 + y^2)} \right) \tag{8}$$

and a radial flow

$$\vec{v}(\vec{r}, t) = \frac{\omega \Delta R R_0^2}{r^2} \cos(\omega t) \vec{e}_r. \tag{9}$$

Force density is computed from the divergence of the stress tensor $f_i = \partial_j \sigma_{ij}^{viscous}$ for the viscosity parameters of 5CB, $\alpha_1 = -0.011$ Pa·s, $\alpha_5 = 0.102$ Pa·s, $\alpha_6 = -0.027$ Pa·s[42]. Other viscosity components do not contribute to a net force in a stationary director field and a radial flow. Due to the director field symmetry, the net force has a

component only in the $z$ direction and equals

$$F_z^{\text{shear}} = \int_{r > R_0} dV\, \partial_j \sigma_{zj}^{\text{viscous}} \approx 0.48\, \pi R_0 \omega \Delta R \alpha_5 \cos(\omega t). \quad (10)$$

We now estimate the viscous force due to reorientation of the director field by considering a point defect moving with a constant velocity $v_{\text{def}}$ in analogy to the two-dimensional case[42]. The director field of a moving hyperbolic defect at small velocities (Er ≪ 1) has the shape of

$$\vec{n}(\vec{r},t) = (x, y, v_{\text{def}}\, t - z)/\sqrt{x^2 + y^2 + (v_{\text{def}} t - z)^2}. \quad (11)$$

Drag force on a moving point defect is estimated from the energy dissipation rate

$$\Sigma = \gamma_1 \int dV\, \dot{\vec{n}}^2 = \gamma_1 v_{\text{def}}^2 \pi^2 R_{\max}, \quad (12)$$

where the integration is performed over a spherical region with radius $R_{\max}$. Taking the defect velocity to be equal to the speed of the bubble surface and estimating the size of the point defect region with $R_{\max} \approx R(t)$, the force can be directly estimated from the dissipation rate[42]

$$F_z^{\text{drag}} = \Sigma/v = c_{\text{def}} R(t) v_{\text{def}} = \pi^2 R(t) \omega \Delta R \gamma_1 \cos(\omega t). \quad (13)$$

Comparing Eq. (10) to Eq. (13), we observe that both mechanisms can produce a force in the direction from the defect towards the colloid. The force due to displacement of the point defect is stronger in magnitude, and we use it in the derivation of the swimming dynamics.

## Slow pulsation dynamics

Here we calculate the dynamics of the spherical bubble and the topological point defect in the slow pulsation regime at Er ≪ 1. In the main text, we introduce an effective dumbbell-like description of the bubble and the defect due to a quadratic potential $\mathcal{F} = \frac{k}{2}(d - \epsilon R)^2$ between them, where $d$ is a bubble surface-to-defect distance with the constant $\epsilon = 0.17$ and $k$ is the effective spring constant[41]. For slow pulsation, the bubble surface-to-defect distance $d = z_B(t) - R(t) - z_{\text{def}}(t)$ equals the equilibrium distance $\epsilon R(t)$, which is proportional to the bubble radius. Here, $z_B$ and $z_{\text{def}}$ are the bubble and the defect positions, respectively. From the sinusoidal oscillation of the bubble radius $R(t) = R_0 + \Delta R \sin(\omega t)$, it follows that

$$z_B(t) - z_{\text{def}}(t) = R_0(\epsilon + 1)(1 + \lambda \sin \omega t), \quad (14)$$

where $\lambda = \Delta R/R$. The spring transmits a force $\vec{F}$ that drives the overdamped motion of the point defect and the bubble:

$$F = -c_{\text{def}} R(t) \dot{z}_{\text{def}}(t) = c_B R(t) \dot{z}_B(t), \quad (15)$$

where $c_{\text{def}}$ and $c_B$ are the drag coefficients for the defect and the bubble, respectively. Combining Eq. (15) and the time derivative of Eq. (14), we can express the propulsion force and the bubble position in the slow pulsation regime as Eqs. (1) and (2), respectively.

## Data availability

The data that support the findings of this study are available within the main text and the Supplementary Information. However, further information can be available from the corresponding author upon request.

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

## Acknowledgements

The authors gratefully acknowledge financial support from the National Research Foundation (NRF) of Korea. S.-J.K. acknowledges NRF-2018R1A6A3A01010921 and IBS-R020-D1. E.U. acknowledges NRF-2022R1A2C1010700. J.J. acknowledges NRF-2020R1A4A1019140 and NRF-2021R1A2C101116312. Ž. K. acknowledges funding from Slovenian Research Agency (ARRS) under contracts P1-0099 and N1-0124. The authors would like to thank Hyuk Kyu Pak and Simon Čopar for valuable discussions and feedback.

## Author contributions

S.-J.K. conceived the idea and performed experiments. Ž.K. developed the theoretical model. E.U. and J.J. designed and supervised the research. S.-J.K., Ž.K., E.U., and J.J. analyzed the data and wrote the manuscript.

## Competing interests

The authors declare no competing interests.
