## [Peer Review File · Nature Communications]

Reviewers' Comments:

Reviewer #1:

Remarks to the Author:

The paper "Centrosymmetric and reciprocal swimmers in a symmetry-broken fluid" is a combined experimental/theoretical study that demonstrates that the nonlinearities inherent to a nematic liquid crystal (due to the geometry of the director) can be used to make new swimmers. Bubbles undergo time-reversible growth/shrinkage and due to point-defects in the fluid they can undergo net motion. The results are new, interesting, and the data is very clean. The paper is mostly nicely written (modulo some weird English sentences) and it was a pleasure to learn about this new physics. There are a few issues though, that I think should be addressed in a revision:

1. The title is quite confusing. I challenge anybody to read this title and figure out what the paper is doing. I think the title should be more specific.

2. The squirmer modelling that is applied is reasonable to me, with a linear response in the tangential direction with a phase delay that leads to swimming. How does that phase delay vary with λ ? Couldn't that contribute to the difference between theory and experiments?

3. The squirmer model is the classical one in a Newtonian fluid. Is it clear that the stresses arising from the microstructure of the LC can be safely neglected? The authors need to demonstrate that this is the case.

4. One important aspect of the study is not clear: what the detailed physical mechanism that leads to the phase delay and therefore to swimming? I believe the data and I believe that this indeed a swimmer, but the authors have not discovered yet the detailed physical mechanism for the motion. What happens to the director field exactly that leads to the phase-delayed tangential motion and thus to swimming? At the very minimum, the paper needs a sketch/cartoon to explain out the details (if the authors understand them!)

Reviewer #2:

Remarks to the Author:

In this manuscript, the authors study experimentally the propulsion of bubbles in a nematic liquid crystal. The pressure of the fluid is modulated periodically and induces pulsations of the bubbles. Even though the bubbles remain centrosymmetric, the local symmetry of their environment is broken by the presence of topological defects in the director field of the liquid crystal, which results in a net displacement of the center of mass of the bubble. The authors first present the conditions under which such propulsion can be achieved. Then, they quantify the displacement of the bubbles, and propose some theoretical arguments to rationalise their observations.

Although the present experimental system could possibly raise some interest in different communities (microswimmers, liquid crystals, active matter), I cannot recommend publication in Nature Communications, for the reasons I give below.

In my opinion, there are many shortcomings regarding the presentation of the data and their analysis, that make several conclusions quite wobbly:

1) The authors estimate power laws on data sets that span a very limited range. For instance, on Figs. 2(b) and 2(c), the x-range is less than a decade, which really questions the reliability of the estimated power law. The x-range on Fig. 2(d) is a bit more extended (1 decade) but still quite small.

2) In my opinion, there are different flaws in the theoretical analysis of the results (starting line 106). First, I do not understand the boundary conditions used by the authors. The authors mention a 'nonslip boundary condition' (line 115), which seems quite inappropriate in the present case given that an air-liquid interface would be better described by a stress-free boundary condition. They also mention 'the slip boundary condition' on line 125. This is not clear, and the authors

should include a sketch of the geometry they consider for their analysis, and I suggest that they write down explicitly the Stokes problem that they solve as well as the chosen boundary conditions.

3) The hypothesis in line 128 is too loosely justified, although the supposed scaling of B_1 with λ is central in the derivation of the power laws in the paragraph right after.

4) The truncation of the expansion over Legendre polynomials (line 131, which bases the rest of the analysis on "no consideration of higher-order terms") is not justified. Relating properly the experimental data to the squirmer model requires a thorough analysis of the data in Fig. 2(e), by extracting the velocity field, projecting it onto the Legendre modes, estimate the coefficients A_i and B_i , in order to justify which modes are negligible and which are not. This analysis is clearly lacking.

5) Therefore, the comparison between the exponents extracted from limited-ranged data sets with a rather unprecise theoretical derivation looks overreaching. The sentence in the abstract ('A squirmer model... explains the underlying swimming mechanism') is an overstatement.

6) Finally, Figs. 2(e) and 3(b) are difficult to read. The authors should plot the flow field as a vector field, with arrows indicating the direction of the flow, and background colors indicating the local magnitude of the velocity. This is a very more standard and readable way to represent flow fields.

Finally, I believe that the authors do not sufficiently justify the potential impact of their experimental system, in a context where many microswimmers have already been designed and studied experimentally. More precisely:

7) The actuation of the bubbles requires some external action (namely a pressure modulation imposed externally) which limits the interest of the model. This should be compared to other artificial microswimmers that do not need any continuous external action to propel themselves, whether they are anisotropic (e.g. Janus particles catalysing chemical reactions) or isotropic (e.g. water droplets propelled by self-sustained Marangoni flows).

8) The interest of considering nematic liquid crystals as an environment for the microswimmers is not sufficient. The authors mention in their conclusion 'porous media and body fluids', which I think are too loosely connected to the present study. Generally, I find the introduction and conclusion too elusive when it comes to the potential impact and interest of the present work.

9) The propulsion velocity typically achieved is extremely small (0.003 body lengths/second according to lines 66-67), which is very low (see for instance Table I in Rev. Mod. Phys. 88, 045006 (2016) for a comparison with other experimental systems).

Reviewer #3:

Remarks to the Author:

The manuscript reports experimental observation of directed motion of pulsating spherical bubble immersed into nematic fluid. Interestingly, the bubble gains net propulsion despite its centrosymmetric shape and time-symmetric size modulation. This intriguing effect, which is prohibited in low Reynolds number Newtonian fluid by the scallop theorem, arises due to anisotropic nature of environment provided by nematic liquid crystal. Namely, the bubble is accompanied by topological defect which may break the symmetry of director configuration. The authors empirically revealed three scaling relations characterizing the directed propulsion of pulsating bubble. Besides, they proposed the way to maximize propulsion efficiency by noting that the average speed takes its maximum value when the bubble diameter is close to the thickness of the nematic cell.

The paper is original, interesting and scientifically valid in its experimental part. However, theoretical understanding proposed in the manuscript is rather questionable. On the one hand, authors correctly point that the pulsating bubble acts as a moving boundary which generate radial flow fields, and the liquid crystal responds to this stimuli and relaxes viscoelastically. On the other hand, instead of trying to reproduce the directed propulsion from the first principles of nematodynamics, they immediately apply the predictions of the squirmer model (see page 6). I would like to note that the basic equations of the squirmer model have been originally derived under an assumption of isotropic Newtonian fluid at low Reynolds number conditions and, thus, the authors should clarify why they believe that the squirmer model is relevant to the experimental situation which is discussed in the present manuscript. Besides, instead of explaining the origin of the tangential flow in the system, the authors just postulate its presence. I cannot recommend publishing the manuscript unless these issues are resolved.

Beside the major questions mentioned above, I also have the following minor comment. To reveal the flow field generated by pulsating bubble the authors seeded nematic with tracer microparticles. Please provide the size of the microparticles and concentration. This would help to justify why these microparticles can be treated as passive inclusions which do not impose back-reaction on the carrier flow field.

We are glad that our manuscript received enthusiastic reviews: “new, interesting, and the data is very clean... a pleasure to learn about this new physics” and “original, interesting and scientifically valid.” However, there was one major concern shared by all three reviewers, *i.e.*, the theoretical understanding of our experiments; specifically, the application of Newtonian squirmer model to the liquid crystal problem with some rough assumptions.

To address this, we worked hard and recruited an expert in nematodynamics, Dr. Žiga Kos, as the co-author. We are now thrilled to present our new model adopting liquid crystal nematodynamics explicitly (not just assuming a squirmer in Newtonian fluids), which elucidates the underlying physics behind the experimental results. Please note that experimental results and analysis remain intact in the revised manuscript.

Here we address all reviewers’ comments one by one in detail, and we attach a revised manuscript incorporating the suggestions, highlighting the changes in orange color. In summary of the major changes, we provide the new theoretical model based on the nematodynamics of anisotropic fluids, replacing the hydrodynamics-based squirmer model in Newtonian fluids. We believe the new model gives a convincing explanation of the underlying physical mechanism. Moreover, addressing reviewers’ criticism, we made a new and specific title and justified the impact of our work by comparing our swimmers with other swimmers in the literature and by specifying the interest in utilizing anisotropic fluids. In addition, we rationalized the limited experimental range in the investigation of scaling behaviors. We ask for reviewers’ understanding that we specified only the line numbers of the text changes (not copy-and-pasting) to shorten the response letter.

Reviewer #1 The paper “Centrosymmetric and reciprocal swimmers in a symmetry-broken fluid” is a combined experimental/theoretical study that demonstrates that the nonlinearities inherent to a nematic liquid crystal (due to the geometry of the director) can be used to make new swimmers. Bubbles undergo time-reversible growth/shrinkage and due to point-defects in the fluid they can undergo net motion. The results are new, interesting, and the data is a very clean. The paper is mostly nicely written (modulo some weird English sentences) and it was a pleasure to learn about this new physics. There are a few issues though, that I think should be addressed in a revision:

1. The title is quite confusing. I challenge anybody to read this title and figure out what the paper is doing. I think the title should be more specific.
 - **Response:** We appreciate your kind suggestion. We have changed the title to “Symmetrically pulsating bubbles swim in an anisotropic fluid by nematodynamics.” In the revision, we developed a new model to elucidate the swimming mechanism of bubbles using the first principles of “nematodynamics” based on the Reviewer comments. Thus, in the new title, we specified our experimental setup — the pulsating bubble in an anisotropic fluid — and summarized the key swimming mechanism, *i.e.*, nematodynamics. We hope the new title helps readers comprehend the paper’s contents easily.
 - **Text changes:**
 - Original title:** Centrosymmetric and reciprocal swimmers in a symmetry-broken fluid
 - Changed title:** [Lines 1–2] Symmetrically pulsating bubbles swim in an anisotropic fluid by nematodynamics
2. The squirmer modelling that is applied is reasonable to me, with a linear response in the tangential direction with a phase delay that leads to swimming. How does that phase delay vary with lambda? Couldn’t that contribute to the difference between theory and experiments?

- **Response:** We appreciate your comments. Before the revision, we thought a hydrodynamics-based squirmer model was good enough to describe the swimming bubble. However, thanks to reviewers' constructive criticism of the squirmer model developed in Newtonian fluids, which is apparently different from our LC, we decided to consider the anisotropic viscoelastic properties of the LC to explain the bubbles' propulsion. The revised manuscript employed nematodynamics to explain the propulsion mechanism.

We found no noticeable correlation between phase delay and λ under the same pulsation frequency using our experimental results, as shown in the newly prepared Supplementary Figure 3. Instead, we again verified that the phase delay depends on the pulsation frequency. Also, the example in lines 103–126 shows how a periodic distortion propagates diffusively through the nematic LC with the effective phase velocity scaling as $\sqrt{\omega}$ but with no explicit scaling with λ . This argument is consistent with our experimental observation of no correlation between phase delay and λ .

- **Text changes:**

Changed text-A: [Lines 103–126]

Changed text-B: [Lines 311–347]

- **Added Figures:** Supplementary Figure 3

3. The squirmer model is the classical one in a Newtonian fluid. Is it clear that the stresses arising from the microstructure of the LC can be safely neglected? The authors need to demonstrate that this is the case.

- **Response:** To address the Reviewer's question and similar comments from other Reviewers, in the revised manuscript, we newly developed a theoretical description based on the nematodynamics' force balance replacing the squirmer model in Newtonian fluids. As the Reviewer pointed out, the main origin of the propulsion force is indeed the nematic microstructure that is periodically deformed by the pulsation of the bubble. The tangential flow, which is the basis of the squirmer-model propulsion, also arises in the new model due to a nonzero average velocity of the swimming bubble.

We included new paragraphs in lines 136 – 195, where we derived the new propulsion model and compared it to experimental results. The theoretical model successfully predicts the experimentally observed scalings in the average propulsion speed and the oscillation amplitude. Furthermore, in the Method (Viscous loss in pulsating flow) (lines 311 – 334), we showed that the new propulsion mechanism is dominant over the propulsion by the anisotropic viscosity of a fixed director field in radial flow.

- **Text changes:**

Changed text-A: [Lines 136–195]

Changed text-B: [Lines 311–347]

4. One important aspect of the study is not clear: what the detailed physical mechanism that leads to the phase delay and therefore to swimming? I believe the data and I believe that this indeed a swimmer, but the authors have not discovered yet the detailed physical mechanism for the motion. What happens to the director field exactly that leads to the phase-delayed tangential motion and thus to swimming?

- **Response:** We are thankful for the Reviewer’s comments. With the new Figure 3a illustrating the bubble and defect as an oscillating dumbbell, we clearly demonstrated that the phase delay is the key factor in the bubble’s propulsion. Furthermore, in here and the revised manuscript, we elucidated the physical mechanism of how the phase delay can arise by the periodic deformation of the anisotropic structure of the nematic LC.

In the revised manuscript, we first examined two possible mechanisms: the anisotropic viscosity of the director field structure and the viscoelastic response of the point defect position. Our calculations show that the latter effect dominates in the oscillation of the bubble position at low frequencies; See Method (Viscous loss in pulsating flow) for the details. The director field around the bubble is in the so-called dipolar structure, where the point defect is at one side of the bubble, and the dipolar moment of the director field scales with the square of the bubble radius (H Stark, Phys. Rep. 351, 387 (2001)). Therefore, the slow pulsation of the bubble radius leads to in-phase periodic changes in the director field structure with the oscillation of the dipolar moment, resulting in the bubble’s oscillation but no net translation. In the slow pulsation, the pulsation period is much longer than the nematic’s relaxation time scale, which means the nematic LC relaxes instantaneously into its equilibrium structure at the pulsating bubble’s radius.

However, at high frequencies, the director field does not instantaneously adapt to changes in the pulsating bubble’s radius, because of the finite relaxation time scale of the nematic. In lines 151 – 195, we brought up one of the analytically solvable examples of the periodically driven nematic, reminiscing our nematic LC around the pulsating bubble. Then, we showed that periodic distortion propagates diffusively through the nematic LC with an effective phase delay. The point defect position and the dipolar moment of the director structure oscillate in time expectedly. But the director profile is no longer in the equilibrium structure for a given radius but modulates with a local phase lag, resulting in the net displacement in the dumbbell model.

- **Text changes:**
Changed text-A: [Lines 151–195]
Changed text-B: [Lines 311–347]

- **Added Figures:** Figure 3a

5. At the very minimum, the paper needs a sketch/cartoon to explain out the details (if the authors understand them!)

- **Response:** We agreed with your comment and added a new cartoon, Figure 3a, to explain the swimming mechanism based on nematodynamics.
- **Added Figures:** Figure 3a

Reviewer #2

In this manuscript, the authors study experimentally the propulsion of bubbles in a nematic liquid crystal. The pressure of the fluid is modulated periodically and induces pulsations of the bubbles. Even though the bubbles remain centrosymmetric, the local symmetry of their environment is broken by the presence of topological defects in the director field of the liquid crystal, which results in a net displacement of the center of mass of the bubble. The authors first present the conditions under which such propulsion can be achieved. Then, they quantify the displacement of the bubbles, and propose some theoretical arguments to rationalise their observations. Although the present experimental system could possibly raise some interest in different communities (microswimmers, liquid crystals, active matter), I cannot recommend publication in Nature Communications, for the reasons I give below. In my opinion, there are many shortcomings regarding the presentation of the data and their analysis, that make several conclusions quite wobbly:

1. The authors estimate power laws on data sets that span a very limited range. For instance, on Figs. 2(b) and 2(c), the x-range is less than a decade, which really questions the reliability of the estimated power law. The x-range on Fig. 2(d) is a bit more extended (1 decade) but still quite small.

- **Response:** We appreciate your careful review. We indeed studied the scaling behavior within the limited range, *i.e.*, approximately over one order of magnitude. In the revised manuscript, we specified this for readers and clarified unavoidable experimental limitations and the system's nature, which are responsible for the limited range.

First, our optical microscopy and the pressure range limit the pulsation ratio λ . If the λ is decreased to expand the range, the bubble's oscillation amplitude and net displacement become smaller than the optical diffraction limit; the typical net displacement after one pulsation cycle with no confinement effect is already sub-micron. On the other hand, pressure modulation approximately ten times higher in amplitude is required to increase λ by the factor of two, considering the inverse relationship between the pressure and the volume = length³; The pressure range covered by our pump was already ± 1 bar.

In a similar vein, the ranges of frequency and bubble size are limited. It is challenging to observe smaller and larger bubbles because the small ones would dissolve quickly into the LC, and the large ones demand a homogeneously aligned thick LC cell of several hundred microns to mm thickness, which is hard to prepare due to the very long relaxation time. In the case of frequency, the pressure pump's response/settling time greater than ~ 100 ms set the maximum frequency ~ 10 Hz. In other words, when the pulsation period becomes comparable to the response/settling time, the pumps cannot generate the sinusoidal pressure modulation with the set pressure-modulation amplitude; λ decreases as we apply higher frequency, as shown in Supplementary Figure 3.

To sum up, we performed our measurements over the maximal range allowed by the system's nature and our experimental setup and explained this limitation in the revised manuscript. Despite the limited range, the theoretical predictions and experimental data show decent agreements.

- **Text changes:** [Lines 294–310]
- **Added Figures:** Supplementary Figure 3

2. In my opinion, there are different flaws in the theoretical analysis of the results (starting line 106). First, I do not understand the boundary conditions used by the authors. The authors mention a 'nonslip boundary condition' (line 115), which seems quite inappropriate in the present case given that an air-liquid interface would be better described by a stress-free boundary condition. They also mention 'the slip boundary condition' on line 125. This is not clear, and the authors should include a sketch of the geometry they consider for their analysis,

- **Response:** We are grateful for the Reviewer's comment on the theoretical analysis. As other Reviewers commonly expressed concerns regarding our theoretical arguments based on the squirmers in Newtonian fluids, we have now developed a new theoretical model based on the principles of nematodynamics of the LC. The new model considers the elasticity-mediated interaction between the topological point defect and the pulsating bubble — an oscillating dumbbell — and the viscoelastic response of the LC around the bubble. Then, the main origin of the propulsion is the pulsation-induced periodic deformation of the nematic microstructure and the resulting phase delay between the pulsation and the nematic microstructure. In the revised manuscript (lines 136–195 and 311–347), the new model replaced the squirmer model.

Because the newly developed model is based on nematodynamics instead of hydrodynamics, we removed the explanation based on the squirmer model and the boundary conditions. Figure 3a provides the sketch of the new model.

- **Text changes:**
Changed text-A: [Lines 136–195]
Changed text-B: [Lines 311–347]
- **Added Figures:** Figure 3a

3. and I suggest that they write down explicitly the Stokes problem that they solve as well as the chosen boundary conditions.

- **Response:** We thank the Reviewer for suggesting how to improve the theoretical analysis of our results. As mentioned in the previous Response, the previous squirmer model with moving boundary became obsolete in the revised manuscript because our new model is now based on nematodynamics, not the Stokes problem. As the Reviewers pointed out, our continuous phase is not Newtonian fluids where you can solve the Stokes problem but liquid crystals with anisotropic viscoelasticity.

However, nematodynamics' dynamic equations for the velocity and the director field, particularly for fast pulsation, can not be solved exactly. Thus we adopt as a guide a solvable example, *i.e.*, the director field dynamics along the z axis, where periodic driving is enforced at $z = 0$ (discussed in the text in lines 103–126). The dynamic equation for the director tilt angle ϕ equals $\gamma_1 \frac{\partial \phi}{\partial t} = K \frac{\partial^2 \phi}{\partial z^2}$, similarly to the Stokes problem, and the phase delay scales as a square root of the driving frequency. We use this example as a motivation to employ a phase delay between the oscillation of the bubble radius and the applied viscoelastic force. We show that the bubble with such a phase delay can have net propulsion reproducing the experimentally observed scalings.

- **Text changes:** [Lines 103–126]

4. The hypothesis in line 128 is too loosely justified, although the supposed scaling of B_1 with

λ is central in the derivation of the power laws in the paragraph right after. The truncation of the expansion over Legendre polynomials (line 131, which bases the rest of the analysis on "no consideration of higher-order terms") is not justified.

- **Response:** We thank the Reviewer for the careful review and agree that the previous truncation needed better justification than just choosing the simplest fluid flow mode for the simplest possible model.

As mentioned in the previous Reponse, our new model based on nematodynamics makes the Legendre polynomial expansion obsolete, and the tangential flow occurs due to the net displacement caused by the interaction between the point defect and pulsating bubble. The new theoretical approach also provides explanations of the experimentally observed scaling laws regarding the λ , *i.e.*, the pulsation ratio of the bubble.

- **Text changes:** [Lines 151–195]

5. Relating properly the experimental data to the squirmer model requires a thorough analysis of the data in Fig. 2(e), by extracting the velocity field, projecting it onto the Legendre modes, estimate the coefficients A_i and B_i , in order to justify which modes are negligible and which are not. This analysis is clearly lacking.

- **Response:** We entirely agree that the analysis procedure suggested by the Reviewer is required for comparing our experimental data with the squirmer model. Since we provide in the revised manuscript a nematodynamics-based model, which is legitimate in liquid crystals, the squirmer model and accompanying analysis, e.g., identifying dominant Legendre mode, are obsolete. However, for the sake of future works by colleagues, we tried our best to share the original experimental data and the PIV analysis despite the complex flow patterns; See Supplementary Figure 5.

- **Added Figure:** Supplementary Figure 5

6. Therefore, the comparison between the exponents extracted from limited-ranged data sets with a rather unprecise theoretical derivation looks overreaching. The sentence in the abstract ('A squirmer model... explains the underlying swimming mechanism') is an overstatement.

- **Response:** We deleted the overstatement in the Abstract. In the previous Response and the revised manuscript, we explained and specified the experimental limitations regarding the 'limited-ranged' data set. Moreover, we replace the incomplete squirmer model with the legitimate nematodynamics-based model. We hope these can deliver our limited but convincing story to the Readers.

- **Text changes:** [Lines 16–22]

7. Finally, Figs. 2(e) and 3(b) are difficult to read. The authors should plot the flow field as a vector field, with arrows indicating the direction of the flow, and background colors indicating the local magnitude of the velocity. This is a very more standard and readable way to represent flow fields.

- **Response:** Since we provide in the revised manuscript a nematodynamics-based model, which is legitimate in liquid crystals, the squirmer model and accompanying analysis, e.g., identifying dominant Legendre mode, are now obsolete and not required for the new model. However, for the sake of future works by colleagues, we provided the flow vector fields through the PIV analysis in Supplementary Figure 5, as suggested by the

Reviewer. For the PIV analysis, we used an iterative PIV plugin of ImageJ with the normalized correlation coefficient algorithm (Template matching method).

- **Added Figure:** Supplementary Figure 5

8. Finally, I believe that the authors do not sufficiently justify the potential impact of their experimental system, in a context where many microswimmers have already been designed and studied experimentally. More precisely: The actuation of the bubbles requires some external action (namely a pressure modulation imposed externally) which limits the interest of the model. This should be compared to other artificial microswimmers that do not need any continuous external action to propel themselves, whether they are anisotropic (e.g. Janus particles catalysing chemical reactions) or isotropic (e.g. water droplets propelled by self-sustained Marangoni flows).

- **Response:** Thank you for allowing us to revisit the importance of our research. Like your comment, there have been numerous studies on various types of microswimmers, including Janus colloids and active emulsions. By listing up previously reported swimmers and comparing them with ours from the perspectives of symmetries and propulsion mechanisms in the Introduction, we justified the impact of our system. As the Reviewer pointed out, our pulsating bubble requires external stimuli and is not fast, which is not desirable for practical applications of microswimmers. However, we hope our readers find and understand how spatio-temporal symmetry breaking can occur in flowing anisotropic fluids solely, *i.e.*, not by carefully designed swimmers, which may lead to other interesting discoveries and applications.

- **Text changes:**

Changed text-A: [Lines 33–41]

Changed text-B: [Lines 198–202]

9. The interest of considering nematic liquid crystals as an environment for the microswimmers is not sufficient. The authors mention in their conclusion 'porous media and body fluids', which I think are too loosely connected to the present study. Generally, I find the introduction and conclusion too elusive when it comes to the potential impact and interest of the present work.

- **Response:** We agree with this comment and improved the Introduction and Conclusion to highlight the impact and interest of this work better. First of all, we removed loosely connected examples in our outlook. Additionally, to name a few interests in considering LC as an environment, we first find that LC's topological aspects break the spatial symmetry of the symmetry swimmers with no additional effort, such as designing asymmetry swimmers. Furthermore, the nematodynamics involving anisotropic viscoelastic properties of liquid crystals provides time-reversal symmetry breaking. In a different vein, the birefringence of the NLC enables us to observe indirectly how the alignment changes due to generated flow patterns.

- **Text changes:**

Changed text-A: [Lines 42–48]

Changed text-B: [Lines 218–226]

Changed text-C: [Lines 256–264]

10. The propulsion velocity typically achieved is extremely small (0.003 body lengths/second according to lines 66-67), which is very low (see for instance Table I in Rev. Mod. Phys. 88,

045006 (2016) for a comparison with other experimental systems).

- **Response:** We agree that the bubble's swimming speed is relatively slow compared to other microswimmers. We provide an honest comparison in the revised manuscript, according to the suggested reference, *Rev. Mod. Phys.* 88, 045006 (2016). Although the confinement effect increases the speed by an order of magnitude, the speed itself is not remarkable. Our focus is that we unveil the novel propulsion mechanism by the spatiotemporal symmetry breaking in anisotropic fluids, which could be small or sometime ignorable.
- **Text changes:** [Lines 198–200]

Reviewer #3 The manuscript reports experimental observation of directed motion of pulsating spherical bubble immersed into nematic fluid. Interestingly, the bubble gains net propulsion despite its centrosymmetric shape and time-symmetric size modulation. This intriguing effect, which is prohibited in low Reynolds number Newtonian fluid by the scallop theorem, arises due to anisotropic nature of environment provided by nematic liquid crystal. Namely, the bubble is accompanied by topological defect which may break the symmetry of director configuration. The authors empirically revealed three scaling relations characterizing the directed propulsion of pulsating bubble. Besides, they proposed the way to maximize propulsion efficiency by noting that the average speed takes its maximum value when the bubble diameter is close to the thickness of the nematic cell. The paper is original, interesting and scientifically valid in its experimental part.

1. However, theoretical understanding proposed in the manuscript is rather questionable. On the one hand, authors correctly point that the pulsating bubble acts as a moving boundary which generate radial flow fields, and the liquid crystal responds to this stimuli and relaxes viscoelastically. On the other hand, instead of trying to reproduce the directed propulsion from the first principles of nematodynamics, they immediately apply the predictions of the squirmer model (see page 6). I would like to note that the basic equations of the squirmer model have been originally derived under an assumption of isotropic Newtonian fluid at low Reynolds number conditions and, thus, the authors should clarify why they believe that the squirmer model is relevant to the experimental situation which is discussed in the present manuscript.

- **Response:** We deeply appreciate the reviewer's suggestion. To derive the directed propulsion from the first principles of nematodynamics instead of relying on the squirmer model in Newtonian fluids. In the revised manuscript, we have taken the suggested approach and proposed a new model. The model employs a dumbbell connecting a defect and a pulsating bubble and studies the nematodynamics around the pulsating bubble. We consider the anisotropic viscosity of the nematic fluid, the viscous drag of moving point defects in the director field, and the elastic interaction between the point defect and the spherical bubble. This model can predict the scaling of the average swimming speed with $\lambda^2 \sin \psi$ and the scaling of oscillation amplitude with λ . We have replaced the text in lines 103–126 and 311–347 to include the derivation and discussion of the new model derived from nematodynamic principles. In lines 136–195, we successfully compare the new model's results to experiments.

- **Text changes:**

- **Changed text-A:** [Lines 103–126]

- **Changed text-B:** [Lines 136–195]

- **Changed text-C:** [Lines 311–347]

- **Added Figure:** Figure 3a

2. Besides, instead of explaining the origin of the tangential flow in the system, the authors just postulate its presence. I cannot recommend publishing the manuscript unless these issues are resolved.

- **Response:** Thank you for your meticulous review. Indeed, postulating the tangential flow was one of the drawbacks of our previous model, i.e., the squirmer model where

the pulsating bubbles behave as squirmers with a specific functional dependence of coefficients $A_i(t)$ and $B_i(t)$. We have now improved the model to consider the propulsion mechanism based on the first principles of nematodynamics; See our Response to your first comment. In this new approach, the tangential flow is a consequence of the nonzero average velocity of the swimming bubble, not a postulate in the model.

3. Beside the major questions mentioned above, I also have the following minor comment. To reveal the flow field generated by pulsating bubble the authors seeded nematic with tracer microparticles. Please provide the size of the microparticles and concentration. This would help to justify why these microparticles can be treated as passive inclusions which do not impose back-reaction on the carrier flow field.

- **Response:**As you suggested, we measured the size and concentration of the tracer particles. We imaged the tracer particles in the isotropic phase of the nematic liquid crystal we used using $60\times$ objective lens and found the average diameter of 12 particles in the field of view was $1.2 \pm 0.3 \mu\text{m}$. To estimate the number density of the tracer particles in a larger field of view, we used $10\times$ objective lens, and the concentration measured from the background image of Supplementary Figure 5d was $2 \times 10^{11} \text{ m}^{-3}$.

The identity of the tracer particles is unclear, and we suppose they are fine dust/impurity particles from air or syringes used. After the bubble injection with a syringe (See Methods for the details), we found a residue of $\sim 1 \mu\text{m}$ could form at the location where a bubble disappeared. Their sizes remained tiny compared to the bubbles, and the number density was low enough not to affect the fluid flow due to the pulsation of the air bubbles. The microparticles rather help us track the fluid flow using the PIV (Particle Image Velocimetry) experiment shown in Supplementary Figure 5.

- **Added Figures:** Supplementary Figure 5

Reviewers' Comments:

Reviewer #1:

Remarks to the Author:

Referee report on revised version of manuscript NCOMMS-21-31716A

I would like to congratulate the authors on their work addressing the comments of the three referees. The revised manuscript reads very nicely, both in terms of motivation/context and in terms of mathematical/physical interpretation. It seems like all of my concerns from the original paper have been addressed, and I congratulate the authors on their nice paper.

I only have query left on the revised manuscript. The author explain mathematically the original for a space-time phase delay in the dynamics of the director field angle ϕ on page 4. This makes intuitive sense, but now of course this means there is a phase delay that has a different value for all values of x . Then in Eq 3 there is a (constant) phase delay Ψ that appears in the Force F_{fast} . How is this phase delay computed? How does it depend on the other parameters of the problem? What the authors describe does make sense, and I do believe it, but I would like to understand the origin of Ψ , in particular since the authors state explicitly "the nematic response to periodic modulation is not instantaneous. Thus the propulsion force by the director fields can experience the phase delay", which implies one is a direct result of the other.

Reviewer #2:

Remarks to the Author:

The authors have made significant changes to the manuscript. In particular, they addressed my main concern about the theoretical analysis they presented in the initial version. They now provide a completely different theoretical approach, which is more suited to their experimental data, and which significantly improves the impact of their results.

I am happy to recommend publication in Nature Communications.

Reviewer #3:

None

Reviewer #1

I would like to congratulate the authors on their work addressing the comments of the three referees. The revised manuscript reads very nicely, both in terms of motivation/context and in terms of mathematical/physical interpretation. It seems like all of my concerns from the original paper have been addressed, and I congratulate the authors on their nice paper.

1. I only have query left on the revised manuscript. The author explain mathematically the original for a space-time phase delay in the dynamics of the director field angle ϕ on page 4. This makes intuitive sense, but now of course this means there is a phase delay that has a different value for all values of x . Then in Eq 3 there is a (constant) phase delay ψ that appears in the Force F_{fast} . How is this phase delay computed? How does it depend on the other parameters of the problem? What the authors describe does make sense, and I do believe it, but I would like to understand the origin of ψ , in particular since the authors state explicitly "the nematic response to periodic modulation is not instantaneous. Thus the propulsion force by the director fields can experience the phase delay", which implies one is a direct result of the other.

- **Response:** We appreciate your careful review of our manuscript. We agree that further clarification is needed on the phase delay ψ in Eq. 3, especially in connecting it with the phase delay calculated in the 1-D model illustrated around the line#113. In the revised manuscript, we reinforced the connection and rationalized the ψ . Here we provide a bit more detailed explanation.

Some insight on the nature of the phase delay is provided by the 1-D example in line #113, illustrating the elasticity-mediated director dynamics and resulting phase-delay phenomena. The actual situation — with the moving boundaries of our pulsating bubble — would be much more complicated. Nevertheless, the model teaches us that there exists a distance-dependent phase delay, *i.e.*, x/ξ in line#117, in the director oscillation which decays out exponentially. The characteristic distance ζ of the exponential decay for the material parameters of 5CB under 4-Hz oscillation equals approximately $4\ \mu\text{m}$. This indicates that the phase-delayed director deformation only in the vicinity of the bubble — including the point defect — should be responsible for the swimming motion, as pointed out in line#125. Particularly, as we discuss in "Viscous loss in pulsating flow" of Methods, the displacement of the point defect plays a dominant role. Thus, a finite ψ with no x dependency in Eq. 3 reflects the *dominant* phase-delayed force to the bubble by the defect, in the coarse-grained model.

When we adopt the phase delay ψ in the F_{fast} force (Eq. 3), we do not assume any specific dependence of the phase delay on the model parameters. The existence of the phase delay suffices to give the net displacement of Eq. 6, regardless of dependencies. As shown in Fig. 2b, we find the phase delay Ψ is proportional to $f^{0.46\pm 0.02}$, which is in line with $\psi \propto \sqrt{\frac{\omega\gamma_1}{K}}$ from the 1-D model, but no strong correlation with the pulsation ratio λ , as shown in Supplementary Fig. 3. The dependence of the phase delay on other material parameters could be tested in future works with different LC materials.

- **Text changes:**

Before: As discussed above for the one-dimensional director dynamics model with an oscillating-director boundary condition at finite Ericksen numbers, the nematic response to periodic modulation is not instantaneous. Thus, the propulsion force by the director

fields can experience the phase delay with respect to $\dot{R}(t)$. To construct a minimal model of bubble propulsion in this fast-pulsation regime, we impose a periodic sinusoidal propulsion force with a phase delay ψ :

$$F_{\text{fast}} = a\omega\lambda R_0^2 \cos(\omega t - \psi), \quad (3)$$

where a is the proportionality coefficient that can depend on the system size and material parameters.

...

After: As discussed above for the one-dimensional director dynamics model with an oscillating-director boundary condition at finite Ericksen numbers, the nematic response to periodic modulation is not instantaneous. Thus, to construct a minimal model of bubble propulsion in this fast-pulsation regime, we employ a phase delay ψ with respect to $\dot{R}(t)$ in the periodic sinusoidal propulsion force:

$$F_{\text{fast}} = a\omega\lambda R_0^2 \cos(\omega t - \psi), \quad (3)$$

where a is the proportionality coefficient that can depend on the system size and material parameters.

...

The origin of the phase delay ψ in Eq. 3, resulting in the net translation of the bubble, deserves further discussion. The one-dimensional director dynamics model explained above shows that the director oscillation decays out exponentially from the boundary. This indicates that the phase-delayed director deformation only in the vicinity of the bubble — including the point defect — matters. Moreover, the force due to a moving point defect is dominant, as shown in Methods. Therefore, in the coarse-grained dumbbell model, we consider the force by the point defect exhibiting the delayed motion with a finite ψ , but do not assume any specific dependence of the phase delay on the model parameters. In Fig. 2(b), experimental data shows that the phase delay Φ between $R(t)$ and $z_B(t)$ scales as $f^{0.46 \pm 0.02}$, which is in line with ψ in the one-dimensional model of director oscillation that is proportional to $\sqrt{f\gamma_1/K}$. The dependence of the phase delay on the other material parameters can be tested in future works using different LC materials.